# Temperate Air Breathing Increases Cycling Performance in Hot and Humid Climate Environment

**DOI:** 10.3390/life11090911

**Published:** 2021-09-01

**Authors:** Clovis Chabert, Aurélie Collado, Olivier Hue

**Affiliations:** Laboratoire ACTES, Université des Antilles, UPRES-EA 3596 UFR-STAPS, 97110 Pointe-à-Pître, Guadeloupe, France; aurelie.collado@univ-antilles.fr

**Keywords:** thermal strains, exercise thermoregulation, respiratory airflow, athletes, thermotolerance, thermal comfort and perception

## Abstract

Practicing physical activity in a hot and humid climate (HHC) is becoming increasingly common due to anthropogenic climate change and the growing number of international sports events held in warm countries. The aim of this study was to understand the physiological and psychological effects of breathing two air temperatures during cycling exercise in HHC. Ten male athletes performed two sessions of exercise in HHC (T°: 32.0 ± 0.5 °C, relative humidity: 78.6 ± 0.7%) during which they breathed hot air (HA, 33.2 ± 0.06 °C) or temperate air (TA, 22.6 ± 0.1 °C). Each session was composed of 30 min of pre-fatigue cycling at constant intensity, followed by a 10 min self-regulated performance. During pre-fatigue, TA induced a better feeling score and a lower rating of perceived effort (respectively, +0.9 ± 0.2, *p* < 0.05; 1.13 ± 0.21; *p* < 0.05) with no changes in physiological parameters. During performance, oxygen consumption and mechanical workload were increased by TA (respectively, +0.23 ± 0.1 L min^−1^, *p* < 0.05 and +19.2 ± 6.1 W, *p* < 0.01), whereas no significant differences were observed for psychological parameters. Reducing the breathed air temperature decreased the discomfort induced by HHC during exercise and increased the performance capacity during self-regulated exercise. Thus, breathed air temperature perception is linked to the hardship of training sessions and directly contributes to the performance decrease in HHC.

## 1. Introduction

A hot and humid climate (HHC), associated with a relatively high temperature and high humidity and commonly associated with a tropical climate, is known to significantly decrease the body’s capacity to eliminate the heat produced by cell metabolism. The high relative humidity (RH) of the HHC prevents the evaporation of sweat, neutralizing its exothermic capacity, and the relatively high temperature considerably reduces the body’s thermal losses by conduction, convection and radiation [1]. Practicing activity/exercise in a HHC induces a performance decrease associated with a higher risk of casualties, such as heat stroke, injury, dehydration, or hyponatremia [2,3,4,5]. In HHC, the mean processes to regulate body temperature are insufficient to compensate the high thermal load associated with the thermogenesis induced by physical activity and thus contribute to an early cessation of activity by practitioners of brief to long-term exercise [6,7]. The sports performance decreases in tropical climates are also affected by cardiovascular adjustments [8], dehydration [3] and elevated skin temperature [9]. However, the limited exercise capacity in a tropical climate is not solely the consequence of physiological repercussions.

The psychological aspect also seems to be closely involved through a disruption in motivation and mood states, as well as through the increased rating of perceived effort and the decreased valorization of the performance by the athletes [7,10]. Furthermore, several studies have shown an anticipatory reduction in the exercise capacity during exposure to heat by the central nervous system (CNS), independently of dehydration or the body temperature increase [11,12].

In order to improve the performance of athletes exposed to a HHC, many countermeasures influencing physiological or psychological factors have been tested [13,14,15,16,17,18]. Among them, acclimation to the tropical climate limits the performance decrease but is still insufficient to completely restore it to a level similar to that obtained in a temperate climate [14,19]. Cooling techniques have also been tested before and during exercise with various levels of success [15,16,20,21]. In the past few years, several authors have focused on the potential use of menthol to counteract the deleterious effect of tropical climates on performance [22,23]. Menthol effects may be due to a disturbance in the perceived temperature, suggesting that the decrease in athletes’ performances under a HHC may be partially due to the elevated temperature perceived in the upper airways. The relation between temperature perception and performance capacity was also explored by Birki et al. [24], who attempted to influence the sensory information perceived by individuals during physical performance through colored environments (i.e., warm vs. cool colors). In this study, the authors showed that the covered distance and heart rate (HR) were significantly lower in the red environment (i.e., a warm color) than in the green environment (i.e., a cool color), evidencing the importance of this perception in performance. Given these findings, as well as the findings suggesting a key role of the oral cavity receptors [25], determining the relative importance of the perception of the inspired air temperature might provide greater insights into the mechanisms of performance limitation in HHC, which in turn might indicate new countermeasures to improve performances in this climate.

The aim of this study was to understand the relation between the temperature perceived in the upper airways and the performance in HHC by characterizing the physiological (ventilatory and cardiac function) and psychological (perceived exertion, mood states, thermal comfort) effects. During a dissociation where the temperature of the inspired air is lower than the environmental temperature (i.e., HHC environment), we found an increased exercise capacity of almost 20 watts linked to a reduction in the perceived exertion and thermal discomfort when cycling intensity is constant (i.e., pre-fatigue).

## 2. Materials and Methods

### 2.1. Population

Ten well-trained male athletes (37 ± 3.3 years old) living in Guadeloupe for at least 5 years and cycling more than three times/week were included (Table 1).

The study was approved by the ethics committee of the Training and Research in Sports Science Unit in Guadeloupe (Ministry of Higher Education and Research). All athletes completed a medical screening questionnaire and gave written informed consent prior to the study, which was conducted according to the Declaration of Helsinki.

### 2.2. Experimental Design

Participation in the study consisted of three sessions of graded exercise testing (GET) and two cycling sessions in different conditions: temperate air breathing (TA) or hot air breathing (HA). The sessions were performed in a randomized manner to avoid the potential effect of habituation or anticipatory changes. They were separated by at least 48 h to avoid fatigue and were performed at the same time of day. The experiments could not be blinded, and the instructions and athletes’ stimulation were therefore standardized for sessions and athletes to reduce the risk of bias. All measurements were performed with an SRM Indoor Trainer electronic cycloergometer (Schoberer Rad Meßtechnik, Jülich, Germany) associated with a Metalyzer 3B gas analyzer system (CORTEX Biophysik GmbH, Leipzig, Germany). Cardiorespiratory parameters were recorded cycle-to-cycle during all testing to obtain HR with a Suunto dual belt and respiratory parameters (VO_2_, respiratory exchange ratio: RER, minute ventilation: V’E) all along the test sessions. All athletes performed a GET protocol to determine their maximal aerobic capacities and maximal aerobic power (MAP), and to confirm their ability to practice exercise (Table 1). Athletes included in the study performed two sessions of cycling composed of a warm-up (5 min at 30% MAP), a pre-fatigue period at a constant mechanical load (30 min at 50% MAP) followed by 10 min of rest, and a performance period (10 min at self-regulated intensity). During the performance period, the athletes managed the brake of the cycloergometer themselves with the instruction to produce the best effort they could during the 10 min of cycling.

All cycling sessions were performed in a HHC environment (T°: 32.0 ± 0.5 °C, RH = 78.6 ± 0.7%; Figure 1a). In this environmental condition, two different breathed air temperatures were tested by conditioning the temperature of the room where the air was collected (Room 2) at 33.2 ± 0.06 °C for the HA condition and at 22.6 ± 0.1 °C for the TA condition (Figure 1 for details).

### 2.3. Measurement of Body Core Temperature and Water Loss

For each session, each subject’s temperature was tracked with an ingestible thermometer (e-celsius, Hérouville Saint-Clair, France). During pre-fatigue, temperature was measured at T-0, T-10, T-20 and T-30 min of cycling, while during the performance test, the temperature was measured at T-0, T-5 and T-10 min of cycling.

The athletes’ water loss during the two sessions was evaluated through the change in body mass. We calculated the ratio of body mass before and after each session and subtracted the water they drank, corresponding to 3× body mass in water (e.g., 210 mL for 70 kg) at 15 min and 55 min of each session. To avoid bias, they were not allowed to use the restroom during the sessions.

### 2.4. Measurement of Psychological Parameters

Before pre-fatigue and the performance test, psychological measurements were performed. Athletes completed three visual analogic scales (VAS) to give their subjective (i) thermal sensation, from −3 (very cold) to 3 (very hot); (ii) comfort, from −2 (very uncomfortable) to 2 (very comfortable); and (iii) acceptability, from −1 (clearly unacceptable) to 1 (clearly acceptable). See Xiong J. et al. for a similar procedure [26]. The athletes completed the feeling scale [27], a VAS ranging from −5 (unpleasurable) to 5 (pleasurable) focusing on subject feelings, several times during pre-fatigue (T-0, T-10, T-20, T-30 min) and performance (T-0, T-5, T-10 min). The perception of exertion ranging from 6 (no exertion) to 20 (maximal exertion) was evaluated by the 15-point Borg’s rating of perceived exertion (RPE; [28]) at T-10, T-20, T-30 min of pre-fatigue and T-5 and T-10 min of performance. To avoid bias, encouragement was given and psychological measurements were made in identical ways across athletes and conditions (HA vs. TA).

### 2.5. Statistical Analysis

All data collected during the study were assessed separately (pre-fatigue and performance session) with a repeated measures ANOVA according to the subject, with condition (HA vs. TA) as the between-factor and period as the within-factor (T-0, T-10, T-20, T-30 for the pre-fatigue period; and T-0, T-5, T-10 for the performance period). Depending on the frequency of data acquisition, we performed statistical analyses on periods (e.g., T-10−T-20 min vs. T-20−T-30 min), which are symbolized with squared brackets in the figures, or time points (e.g., T-10 min vs. T-20 min). If a dataset did not fit the normality law (Shapiro-Wilk test) and homogeneity characteristics (Levene test), the data were corrected with Box-Cox transformation. Holm-Sidak post-hoc tests were performed to determine the detected effects with a α threshold at *p* = 0.05. Feeling Scale data were analyzed with Friedman’s ANOVA with a Bonferroni correction of the *p*-value (threshold at *p* = 0.05) and the effect size was calculated with Kendall’s W (coefficient of concordance).

## 3. Results

### 3.1. Consequences of Air Temperature on the Athletes’ Mechanical Power

The athletes’ mechanical power was recorded during all pre-fatigue and performance sessions in both tested conditions (HA vs. TA).

During pre-fatigue, the cycling power was maintained at 50% MAP by computer, which explains the absence of difference observed between HA and TA conditions (respectively, 192.3 ± 8.1 W vs. 191.3 ± 7.0 W; ns. Figure 2a). However, performance was 19.2 W higher when athletes were exposed to TA compared to HA (respectively, 245.4 ± 16.6 W and 226.2 ± 12.6 W, *p* < 0.01, η^2^_p_: 0.56. Figure 2b). In both conditions, performance was completed at a higher intensity than the pre-fatigue intensity (respectively, 235.8 ± 10.4 W and 191.8 ± 5.2 W, *p* < 0.01, η^2^_p_: 0.66), corresponding to 59% for HA and 64% for TA of the MAP measured during GET.

### 3.2. Follow-Up of the Thermoregulatory Process via Evolution of Body Temperature and Body Mass

The measurements of central temperature presented in Figure 3a were not influenced by air inlet temperature during the pre-fatigue cycling. However, a global linear increase in the central temperature was observed all along the sessions from 37.04 ± 0.1 °C to 38.4 ± 0.1 °C (*p* < 0.001, η^2^_p_: 0.91). As observed for pre-fatigue, performance induced a global increase in central temperature (*p* < 0.001, η^2^_p_: 0.55) at T-10, which was different from rest and T-5 (respectively, +0.45 ± 0.09 °C; *p* < 0.001 and +0.14 ± 0.07 °C; *p* < 0.05). Body mass loss presented in Figure 3c was not changed by the air inlet temperature (HA vs. TA; *ns*). As the athletes had ingested no food, this weight loss was due to substrate oxidation, which is negligible for a 45 min cycling period, and water loss by sweating. Thus, in the HA and TA inlet conditions, no difference in the sweating rate was observed (respectively, 1.73 ± 0.21 kg and 1.71 ± 0.22 kg; *ns*).

### 3.3. Evolution of Physiological and Psychological Parameters during Pre-Fatigue

As expected, the O_2_ consumption presented in Figure 4a increased in both HA and TA conditions (*p* < 0.001, η^2^_p_: 0.97) from T-10 min (respectively, +2.22 ± 0.13 L min^−1^ and +2.08 ± 0.12 L min^−1^; *p* < 0.001) and was maintained relatively constant during all the pre-fatigue. V’E (Figure 4c) followed the same variation profile with an increase induced by cycling (HA: +58.0 ± 4.7 L min^−1^ and TA: +54.5 ± 5.2 L min^−1^; *p* < 0.001, η^2^_p_: 0.94), which was maintained all along pre-fatigue. Heart rate increased with exercise (*p* < 0.001, η^2^_p_: 0.99) in HA and TA conditions (respectively, +69.0 ± 3.3 beat min^−1^ and +69.2 ± 2.8 beat min^−1^; *p* < 0.001), and a cardiac drift was observed at T-30, which was significantly higher than at T-10 (HA: +17.5 ± 0.4 beat min^−1^ and TA: +15 ± 1.4 beat min^−1^; *p* < 0.001) and T-20 (HA: +8.4 ± 0.8 beat min^−1^ and TA: +6.8 ± 1.0 beat min^−1^; *p* < 0.001). During pre-fatigue, no significant variations in RER were observed in either the HA or TA condition.

Contrary to the cardiorespiratory parameters, the psychological parameters presented in Figure 4e,f appeared to be altered by both the inlet air conditions and the cycling task during pre-fatigue. Indeed, the feeling scale (Figure 4e) scores were decreased in the course of time (*p* < 0.001, W: 0.93) in both conditions compared to the rest measures at T-20 (HA: −1.4 ± 0.4 and TA: −1.8 ± 0.5; *p* < 0.05) and T-30 (HA: −2.9 ± 0.6 and TA: −3.1 ± 0.5; *p* < 0.001). Interestingly, the athletes had more pleasurable feelings in TA than HA during all pre-fatigue, despite the same room temperature and absolute cycling intensity (mean feeling score of TA vs. HA: +0.9 ± 0.2; *p* < 0.05, W: 0.30). Figure 4f shows similar results with (i) a global effect of the time factor (*p* < 0.001, W: 0.54) corresponding to an increase in the perceived effort in both conditions at T-20 and T-30 (respectively, when compared to T-10: +0.83 ± 0.31 and +1.2 ± 0.24; *p* < 0.05) and (ii) a lower RPE in TA than HA (1.13 ± 0.21; *p* < 0.05, η^2^_p_: 0.75). However, TA did not have any effect on the thermal sensation, comfort or acceptability items of the VAS completed by the athletes during pre-fatigue (Appendix A).

### 3.4. Evolution of Physiological and Psychological Parameters during Performance

Analysis of the cardiorespiratory parameters during performance (Figure 5) showed patterns of variation similar to those of pre-fatigue cycling concerning VO_2_, HR and V’E.

Indeed, oxygen consumption of HR and V’E (Figure 5a–d) were significantly increased by the cycling exercise (respectively, *p* < 0.001; η^2^_p_: 0.96; η^2^_p_: 0.94; η^2^_p_: 0.97), which is in accordance with current knowledge. The significant differences observed between T-5 and T-10 for HR and V’E (respectively, *p* < 0.001 and *p* < 0.01) may have been the consequence of the final sprint performed by the athletes in the last minutes of the performance cycling period. Indeed, to reach their best effort possible, all the athletes voluntarily increased the mechanical workload they produced during the last moments of the performance. Whereas no differences appeared between the HA and TA conditions during pre-fatigue, the O_2_ consumption was significantly higher with TA than HA at T-5 and T-10 (respectively, +0.23 ± 0.1 L min^−1^ and +0.22 ± 0.09 L min^−1^; *p* < 0.05. Figure 5a), with V’E more increased at T-10 during performance in TA compared to HA (+12.6 ± 5.6 L min^−1^; *p* < 0.01; Figure 5c). Interestingly, the VO_2_ increase in TA (+8.2 ± 2.1%) matched the mechanical workload increase in this cycling condition (+8.2 ± 2.3%).

As seen for the cardiorespiratory parameters, the consequences of cycling during performance on the psychological parameters were similar to those obtained during pre-fatigue. Nonetheless, the impact of HA and TA on these parameters significantly differed from pre-fatigue. Indeed, as presented in Figure 5e, cycling induced a global decrease (*p* < 0.001; W: 0.90) of the athletes’ wellbeing feelings (i.e., an increase in unpleasurable feelings) at T-5 and T-10 when compared to rest period (respectively, −2.7 ± 0.3; *p* < 0.01 and −4.6 ± 0.5; *p* < 0.001). The RPE (Figure 5f) also showed a global increase at T-10 when compared to T-5 (+2.1 ± 0.3; *p* < 0.01; η^2^_p_: 0.61). However, contrary to pre-fatigue, neither of these parameters was influenced by TA during performance. Concerning the athletes’ thermal perception of the environment (Appendix A), TA did not change any of the parameters measured before the performance test, which was similar to the results obtained before pre-fatigue (Appendix A).

## 4. Discussion

The aim of this study was to understand the relation between the temperature perceived in the upper airways and performance in HHC by dissociating the temperature of breathed air from the environmental temperature during a cycling task. The results showed that decreasing the air inlet temperature during a cycling exercise in HHC did not induce physiological variation during the pre-fatigue task, but it resulted in an improvement in the athletes’ mechanical work by almost 20 watts during the 10 min performance. The explanation for this result may involve several physiological or psychological parameters that we will discuss further.

The analysis of the physiological parameters showed that the progressive increase in core temperatures was the consequence of the thermal load imposed on the athletes by the HHC [13] during sports exercise, validating the environmental conditions chosen for this study. The comparison between the TA and HA sessions showed no effect of the breathed air conditions on these two temperature parameters. This absence of difference during pre-fatigue, whose mechanical workload was identical for TA and HA, suggested that the temperate breathed air did not decrease the thermal load on the athletes. This is in accordance with the current knowledge that places heat dissipation by breathing as an anecdotic process [29]. The absence of difference in the total water loss between HA and TA suggests that the sweating rate did not change with the temperature of the breathed air. This may confirm that the thermal load was not significantly affected by the HA or TA condition. The absence of an effect on the cardiorespiratory parameters recorded during pre-fatigue tended to confirm that the air inlet condition played no role in the athletes’ thermal load. An elevated temperature environment is well-known to increase cardiac drift during endurance exercise [30]. The substantial cardiac drift observed in both TA and HA during pre-fatigue despite the low cycling intensity (50% of MAP) may be the consequence of the heat stress undergone by our athletes. During performance, the higher level of oxygen consumption and V’E observed with TA could easily be explained by the increase in the mechanical workload measured in this air inlet condition. Interestingly, despite the respiratory and intensity increases, no difference in HR parameters was found between HA and TA. Dehydration is well-known to induce an HR increase [31], but the athletes’ weight loss was similar in TA and HA. Higher peripheral resistances that lead to an increase in the heart workload [31,32] seems unlikely because no effects were observed in liquid loss and body temperature increases between the breathed air conditions. Thus, the +19.2 W increase during performance could be the consequence of a thermal load decrease induced by TA when the ventilation flow was high. However, psychological parameters may also have contributed to the alterations observed during physical performance in the HHC environment.

Although the mask used to change the breathed air temperature did not allow us to exclude the possibility that exposure of the facial skin to temperate air also contributed to the observed effects, the assessments of the perceived effort and pleasurable feelings (i.e., feeling scale score) with TA during pre-fatigue reduced the exercise discomfort that is known to be associated with the HHC environment. This better tolerance of HHC may have mechanisms in common with the increased performance induced by menthol beverages that dupe athletes’ temperature perceptions by producing a fresh feeling in the mouth [22,23,33]. The better RPE and feeling score measured during pre-fatigue with TA may have contributed to the increased performance by saving the athletes’ CNS resources. This hypothesis is in accordance with a previous study that showed exacerbated CNS fatigue during exercise in the heat that led to an alteration in muscle recruitment and power output [34], although these effects are still under debate [35]. Interestingly, the differences between TA and HA in pleasurable feelings and perceived effort scores remained constant over time for all the pre-fatigue exercise, indicating that the modifications in these parameters by TA were linked only to environment perceptions and not to fatigue related to physical exercise. Thus, TA may reduce the environmental thermal load perceived by athletes without reducing the mechanical load perception caused by the appearance of physiological fatigue. During performance, the differences in the pleasurable feelings and RPE between TA and HA totally disappeared. These results may indicate that each athlete may have reached a threshold for tolerating unpleasurable feelings, as well as a threshold for tolerating the difficulty of a perceived effort, beyond which increases in the mechanical power output became impossible. This hypothesis seems to offer at least a partial explanation for the better mechanical power observed during performance. In this case, the decrease in the thermal load perceived by the athletes in the TA condition enabled them to reach their tolerance threshold at a higher mechanical workload. The results suggest that the use of the RPE to quantify exercise intensity, as described in current literature [36], has to be weighed according to the environmental conditions. Scores associated with the Profile of Mood States questionnaire (Appendix A) did not seem to be sensitive to the breathed air temperature, but this result may have been due to the design of the study. Indeed, the questionnaires were completed at the end of the entire session, whereas the absolute mechanical workload differed between the HA and TA conditions during the performance task. The tight relation between HA and TA for the mood state scores tends to confirm the hypothesis that each athlete had a threshold of perceived disturbance in HHC beyond which they were not able to maintain exercise intensity.

Thus, the main result of this study was the increase in cycling capacity during a 10 min self-regulated performance when the temperature of the breathed air was artificially decreased. In line with the study design and the hypothesis that an unpleasurable feeling threshold contributed to the performance decrease in HHC, the results suggest that modifying the breathed air temperature may be useful for high-intensity aerobic exercise. Nonetheless, further experiments with variations in the cycling task order (pre-fatigue and performance) or a mix of the breathed air conditions between the two cycling tasks (pre-fatigue in TA then performance in HA) may bring new perspectives and applications to this work.

## 5. Conclusions

The original system used in this study to dissociate the breathed air from the environmental temperature in the exercise room contributed to enhancing our understanding of the adverse physiological and psychological outcomes in the HHC. Our results suggest that the decrease in performance in HHC is partially driven by oropharyngeal temperature, which influences feeling perception and the rating of perceived effort. This study highlights the importance of psychological parameters in HHC performance, suggesting temperature perception as a lever to develop better countermeasures for exercise in the heat. That could be an asset in the field of sports performance but also in the field of public health in efforts to help the public better tolerate heat exposure.

## Figures and Tables

**Figure 1 life-11-00911-f001:**
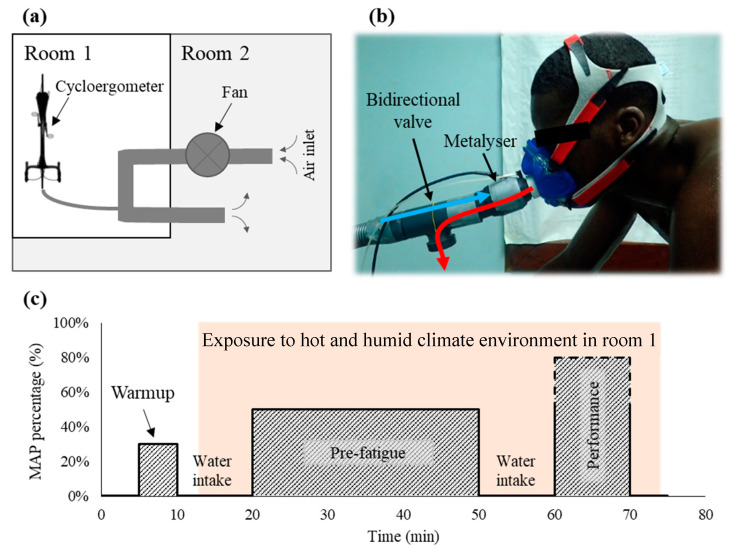
Schematic view of (**a**) the air conditioner used in the study and (**b**) picture illustrating the bidirectional valve used with the inspired air (blue arrow) and the exhaled air (red arrow). Graphical representation of the cycling protocol performed by athletes (**c**) during hot air breathing (HA) and temperate air breathing (TA) sessions. MAP: maximal aerobic power.

**Figure 2 life-11-00911-f002:**
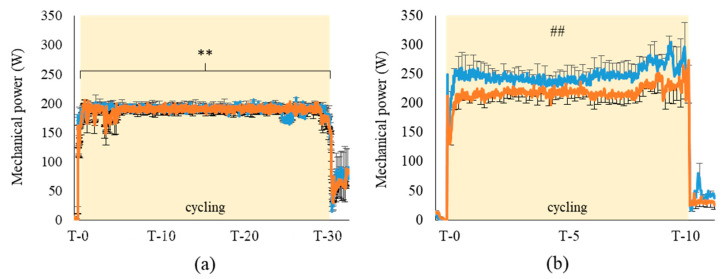
Mean mechanical power provided by athletes during pre-fatigue (**a**; 30 min at 50% MAP) and performance (**b**; 10 min at self-regulated intensity). (*n* = 10; mean ± SEM). ■: hot air breathing (33 °C); ■: temperate air breathing (23 °C). MAP: maximal aerobic power. *: pre-fatigue different from performance (**: *p* < 0.01); #: TA_performance_ diff. from HA_performance_ (##: *p* < 0.01).

**Figure 3 life-11-00911-f003:**
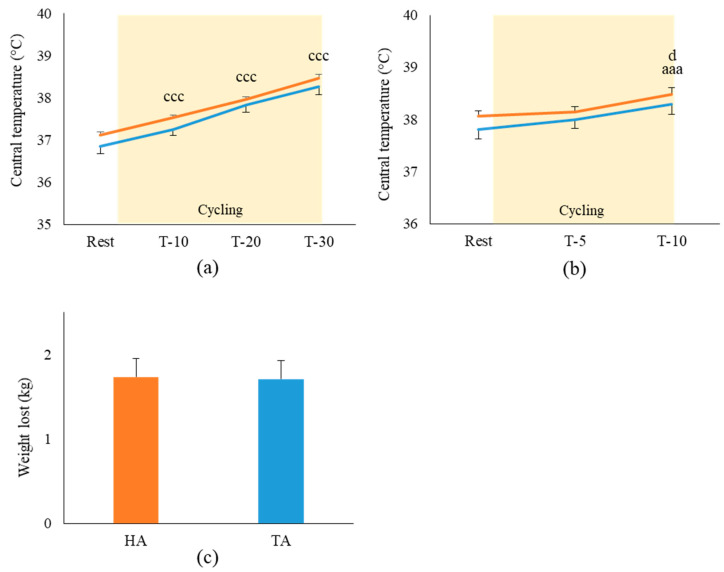
Evolution of central temperature during (**a**) pre-fatigue and (**b**) performance; (**c**) mean body mass loss by athletes during the entire cycling session in each condition. (*n* = 10; mean ± SEM). ■: hot air breathing (HA; 33 °C); ■: temperate air breathing (TA; 23 °C). a: diff. from rest (a: *p* < 0.05; aa: *p* < 0.01; aaa: *p* < 0.001). c: diff. from all recording times (c: *p* < 0.05; cc: *p* < 0.01; ccc: *p* < 0.001); d: T-10 diff. from T-5 (d: *p* < 0.05).

**Figure 4 life-11-00911-f004:**
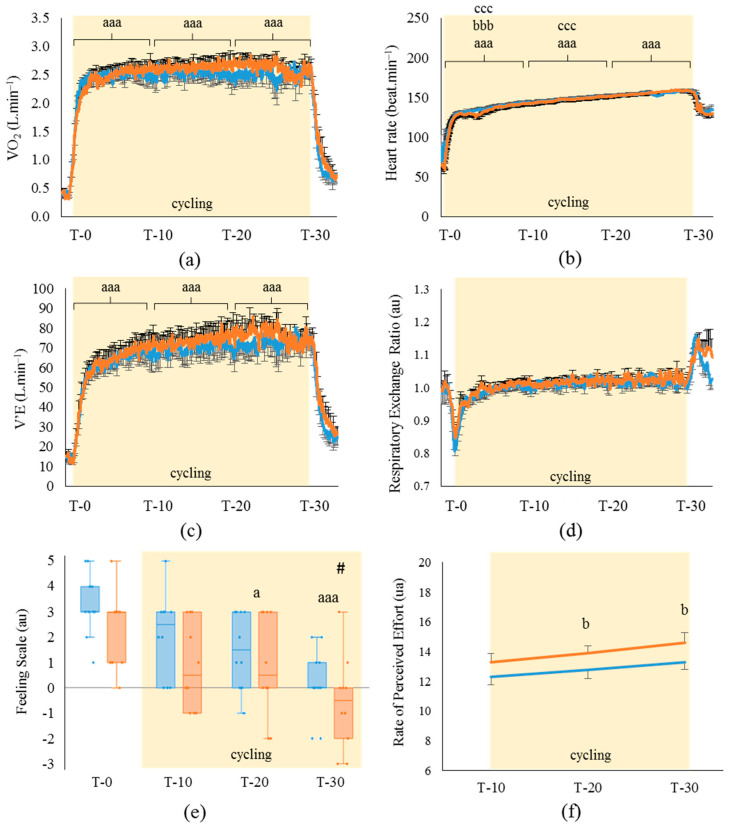
Evolution of (**a**) oxygen consumption, (**b**) heart rate, (**c**) ventilatory flow and (**d**) respiratory exchange ratio during the pre-fatigue session of cycling at 50% of maximal aerobic power for 30 min. Graphical representation of the visual analogic scores for (**e**) the feeling scale and (**f**) the rating of perceived effort scale as perceived by the athletes all along the pre-fatigue cycling. (*n* = 10; mean ± SEM). ■: hot air breathing (33 °C); ■: temperate air breathing (23 °C).V’E: respiratory minute volume; VAS: visual analogic scale. a: diff. from rest (a: *p* < 0.05; aa: *p* < 0.01; aaa: *p* < 0.001); b: diff. from T-30 (b: *p* < 0.05; bb: *p* < 0.01; bbb: *p* < 0.001); c: diff. from T-10 (ccc: *p* < 0.001); #: HA diff. from TA (*p* < 0.05).

**Figure 5 life-11-00911-f005:**
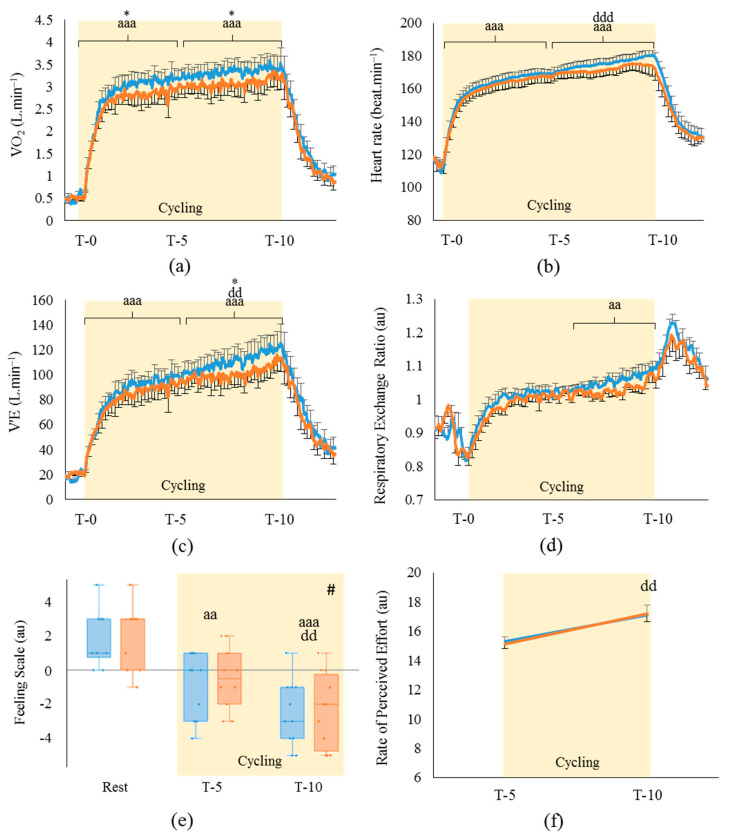
Evolution of (**a**) oxygen consumption, (**b**) heart rate, (**c**) ventilatory flow and (**d**) respiratory exchange ratio during the 10 min cycling performance at self-regulated intensity. Graphical representation of the visual analogic scores on (**e**) the feeling scale and (**f**) the rating of perceived effort by the athletes all along the performance cycling. (*n* = 9−10; mean ± SEM). ■: hot air breathing (33 °C); ■: temperate air breathing (23 °C). V’E: respiratory minute volume. VAS: visual analogic scale. a: diff. from Rest (aa: *p* < 0.01; aaa: *p* < 0.001); d: T-10 diff. from T-5 (dd: *p* < 0.01; ddd: *p* < 0.001); *: hot air diff. from temperate air at measurement time (*: *p* < 0.05; **: *p* < 0.01); #: HA diff. from TA (*p* < 0.05).

**Table 1 life-11-00911-t001:** Morphological characteristics and main outcomes of the graded exercise test performed by athletes (*n* = 10; mean ± SEM). BMI: body mass index; FFM; fat-free mass; BFM: body fat mass; MAP: maximal aerobic power; HRmax: maximal heart rate.

**Morphology**		**Mean**	**±**	**SEM**
Height	(cm)	181.7	±	2.60
Body mass	(kg)	73.04	±	2.7
BMI	(au)	22.04	±	0.56
FFM	(kg)	66.5	±	2.4
BFM	(kg)	8.49	±	0.7
**Graded Exercise Testing**		**Mean**	**±**	**SEM**
VO_2_max	(mL·min^−1^ kg^−1^)	51.77	±	1.71
MAP	(watts)	376	±	11.59
HRmax	(beat min^−1^)	177.7	±	1.5

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
