# Peer review of "Temperate Air Breathing Increases Cycling Performance in Hot and Humid Climate Environment"

_life, 2021, doi:10.3390/life11090911_

Round 1

Reviewer 1 Report

The authors aim to understand how factors related to the temperature of the air breathed during exercise in the heat affects psychological and performance aspects of cycling. Although the main findings sound interesting, the manuscript needs a thorough and complete English grammar revision before it can be re-considered. There are so many typos and mistakes that make the manuscript extremely hard to read and interpret. All supplementary tables and figures mentioned throughout the manuscript should be included in the main article instead of being presented as supplement. In addition, several aspects require clarification. Please find my main concerns below.

Abstract

  • Line 21 – should be “linked” rather than “link”

Introduction

  • What do you mean by “thermolysis capacity of the body”?
  • Please define a temperature range for tropical climate. According to Wikipedia “Tropical climates are characterized by monthly average temperatures of 18 ℃ (64.4 ℉) or higher year-round and feature hot temperatures.” Your study environment is 32C with a humidity of 78%. I recommend you refer to your environmental condition as hot and humid rather than referring to it as tropical climate.
  • Line 26-27 should be “high humidity” rather than “high hygrometry”
  • Line 29 – should be “evaporation of sweat” rather than “vaporization”
  • Authors must emphasize the rationale for the study.
  • Your description of the study [24] in your introduction does not make any sense. Please revise.

Materials and Methods

  • Conventionally TC refers to core temperature in the literature. Your use of TC for tropical climate makes it very confusing to read. Please fix it.
  • Line 90 and 91 - The authors mention that the 10 min Performance period was at maximum intensity (line 90) and then (line 91) that the intensity was self-selected by the subjects. This is inconsistent, please clarify.
  • The authors do not mention the volunteers' cardiac monitoring procedure (device/moments). Authors should add this information in this section.
  • Line 94 - The method of exposure to hot and cold air is unclear. The authors report that all sessions were held in a room with a temperature of 32°C (room 1) and then mention the entry of air into another room (room 2). Authors should rewrite for ease of understanding.

Results

  • Line 134 – Description figure 1: the authors mention that the Pre-Fatigue period was performed at an intensity of 60%, however in the methods it was described that it would be at 50% of the MAP. It is also mentioned that the Performance test was at maximum intensity. Was it maximal intensity or self selected? How to ensure that it was at maximum intensity? Which parameter?
  • Line 138 - The authors reaffirm that in Pre-Fatigue the intensity was 50% MAP. Figure 1 (line 134) mentions 60% MAP.
  • Line 147 - The authors reaffirm that in Pre-Fatigue the intensity was 50% MAP. Figure 1 (line 150) mentions 50% MAP.
  • Standardize the description of hot air and cold air in figures 1, 2 and 3 (figure 1 shows the acronym; figures 2 and 3 show the temperatures for each situation).
  • MAP values ​​in the Performance test. Do authors have? The authors could use this data to demonstrate the intensity achieved in the Performance test.
  • Line 142 - The authors compare the Pre-Fatigue moment (controlled from the MAP percentage) with the Performance moment (without MAP data reached) to demonstrate that the load was higher in the Performance test. It would be interesting to demonstrate the difference achieved, comparing these two moments, with the MAP.
  • The understanding of some results was hampered by the lack of supplementary graphics. These data must be added.

Discussion

  • Line 222 - The “consequences of air temperature on mechanical power provided by athletes” needs to be discussed.
  • Line 241 – The authors rely on a study (reference 29) to explain a hypothesis, however the aforementioned study does not support the information due to the lack of a direct relationship between the parameters. The study mentioned did not assess the temperature of the air being breathed, but the environmental condition. Authors should review this information.
  • Line 246 – The authors discuss dehydration, but there are no results on this variable, which does not allow us to deduce what is stated in the sentence of the line in question.
  • Line 256 - "This better tolerance of TC, induced by a stimulus associated with a temperate environment, may have common mechanisms for those induced by menthol and its effects on performance in a hot environment". What mechanisms? It is interesting that the authors cite them to clarify the relationship between the use of menthol and breathing cold air.
  • What are the main results that the authors consider, Pre-Fatigue or Performance? What is the practical application?

Conclusion

  • The authors mention the original system that dissociated the breathing air from the ambient air, but did not detail the materials and methods. It is important to describe the system in materials and methods. This is critical for the study, please present as figure in the body of the manuscript and not as a supplement.
  • Authors should clarify for their conclusion the possible positive effects of inspired air temperature on psychological parameters of exercise in heat. At what intensity/moment would these benefits be noticeable. In which circumstances related to exercise intensity could we consider the results of the study (mild/moderate as in Pre-Fatigue or high intensity as in the Performance test? The information that will be added in the previous considerations will help in the conclusion.

Reviewer 2 Report

The authors show that breathing temperate air in a hot and humid environment has beneficial effects on certain psychological (but not physiological) parameters during pre-fatigue cycling, whereas it increases the mechanical workload and cardiorespiratory parameters (but does not meaningfully change psychological indicators) during performance cycling. The topic and the results can be of high interest, however several methodological details should be clarified, interpretation and discussion of the results should be improved, and the language and grammar of the manuscript should be thoroughly revised.

MAJOR COMMENTS:

1) In Materials and Methods, (a) the permit/registration number for the approval of the study must be included. (b) It is not described whether the 2 cycling sessions in different conditions (ln. 79) were performed in a randomized manner. This must be described. If the sequence of the sessions was not randomized, then the potential effect of habituation and anticipatory changes should be discussed, which is a limitation of the study. Alternatively, the authors may consider conducting the experiments in a randomized manner. (c) It could be also important that the investigator is blinded to actual intervention (i.e., hot or temperate air breathing). Any attempts for blinding should be also described.

2) Description and citation of Supplementary figures S2 and S3, as well as that of table S2 should be included in the results section, then the findings can be explained in discussion. In the current version supplementary figure S4 is mentioned earlier in the text than S2 and S3. If needed the order of the supplementary figures should be changed to follow the order of their mentioning in the main text. It should be also explained why the authors thought that it was necessary to use 2 different body temperature measurement methods.

3) The authors aimed at understanding the role of the temperature perceived by upper airway in the study, but they used a face mask, which also covered a decent part of the face skin. It cannot be excluded that the exposure of the facial skin to temperate air also contributed to the observed effects. This possibility must be discussed as a limitation of the study.

4) It is not clear how the psychological changes (without any physiological effects) during the pre-fatigue session could affect mechanical power during the performance session (ln. 271-273). Hypothetical explanations should be discussed. The most convincing evidence to confirm or reject an existing relationship between the two would be to run additional experiments, in which the pre-fatigue and performance sessions are performed in a randomized order.

MINOR COMMENTS

1) Ln. 34: Not only thermogenesis due to physical activity should be counteracted, but also the heat load coming from the warm environment. The sentence should be corrected.

2) Instead of the expressions "warm/hot temperatures", the authors should consider to use, for example, high/increased/elevated temperature (e.g., ln. 54-55).

3) Ln. 134 and 150: In the figure legend, “60% MAP” should be corrected to “50% MAP” according to the text of the manuscript.

4) Ln. 138: Why was the absence of a difference expected? If the statement is correct, references should be provided.

5) Ln. 162-168: (a) The sentence should be rephrased, because now it suggests that the difference between 10 and 30 minutes was smaller than between 20 and 30 minutes, which is probably not the case. (b) The symbols (bbb and ccc) in Figure 2 should be also revised as it is not clear whether single time points (e.g., 0 min vs 10 min) or time periods (e.g., 0-10 min vs 10-20 min) were compared with each other. (c) In ln. 162, “measurement” should be deleted from “heart rate measurement increased”. (d) In the unit of heart rate, “bat” should be corrected to “beat” throughout the text. (e) It should be also mentioned that in Figure 2d, RER increases from ~0.8 at T-0 to 1.0 at T-10-30. A similar increase can be seen in Figure 3d, which is also ignored by the authors.

6) Ln. 189-190: Was the “final sprint” obligatory or voluntary? It should be explained here or in the Methods.

7) Ln. 247-250: The sentence does not make sense to me. It should be deleted or rephrased.

8) Supplementary ln. 42-44: It is probably missing from the sentence that the statement is related to performance session.

9) Throughout the “Total body weight loss” section, (a) “bodyweight” should be corrected to “body weight”. (b) Decimal commas in the figure should be changed to decimal points. (c) In the figure legend water intake is mentioned and reference to Methods is given, but there did not find description about water intake in methods. I do not understand what the authors mean on “3x bodyweight in water twice during the session”. This would mean 3 times 70 kg in water twice during 70 minutes. This should be revised. (d) “Transpiration” is a term commonly used in botany, the authors should consider to replace it with, for example, sweating. (e) It should be stated if the subjects were allowed to use the restroom during the experiment as urination and defecation could also contribute to the body weight loss.

10) Supplementary ln. 85: According to the figure legend, the tests were performed before the sessions, which does not make sense. It should be corrected.

11) The use of abbreviations should be revised throughout the manuscript. Some abbreviations are introduced but then the expression is spelled out instead of using the abbreviation (e.g., ln. 150-151: maximal aerobic power and visual analog scale should read as MAP and VAS, respectively; ln. 286 and 287: tropical climate should be TC). This also means that the abbreviation does not need to be introduced again in the figure legend (ln. 136 and 153).

Round 2

Reviewer 2 Report

The authors responded to my comments in a satisfactory manner.

Author Response

thank you